# Phenology Patterns and Postfire Vegetation Regeneration in the Chiquitania Region of Bolivia Using Sentinel-2



**Oswaldo Maillard** [1,*] **, Marcio Flores-Valencia** [1] **, Gilka Michme** [1] **, Roger Coronado** [1] **, Mercedes Bachfischer** [2] **, Huascar Azurduy** [1] **, Roberto Vides-Almonacid** [1] **, Reinaldo Flores** [1] **, Sixto Angulo** [1] **and Nicolas Mielich** [1]

[1] Fundación para la Conservación del Bosque Chiquitano (FCBC), Av. Ibérica Calle 6 Oeste 95, esq. Puerto Busch, Las Palmas, Santa Cruz, Bolivia; mflores@fcbc.org.bo (M.F.-V.); gmichme@fcbc.org.bo (G.M.); rcoronado@fcbc.org.bo (R.C.); hazurduy@fcbc.org.bo (H.A.); robertovides@fcbc.org.bo (R.V.-A.); rflores@fcbc.org.bo (R.F.); sixto@fcbc.org.bo (S.A.); nicolas.mielich@giz.de (N.M.)
[2] The Emergency Program (TEP), 08460 Barcelona, Spain; mercedes@emergprogram.com
* Correspondence: omaillard@fcbc.org.bo or hylopezus@gmail.com

**Abstract:** The natural regeneration of ecosystems impacted by fires is a high priority in Bolivia, and represents one of the country's greatest environmental challenges. With the abundance of spatial data and access to improved technologies, it is critical to provide an effective method of analysis to evaluate changes in land use in the face of the global need to understand the dynamics of vegetation in regeneration processes. In this context, we evaluated the dynamics of natural regeneration through phenological patterns by measuring the maximal and minimal spectral thresholds at four fire-impacted sites in Chiquitania in 2019 and 2020, and compared them with unburned areas using harmonic fitted values of the Normalized Difference Vegetation Index (NDVI) and the Normalized Burn Ratio (NBR). We used two-way ANOVA test to evaluate the significant differences in the values of the profiles of NDVI and NBR indices. We quantified severity at the four study sites using the dNBR obtained from the difference between pre- and postfire NBR. Additionally, we selected 66 sampling sites to apply the Composite Burn Index (CBI) methodology. Our results indicate that NBR is the most reliable index for interannual comparisons and determining changes in the phenological pattern, which allow for the detection of postfire regeneration. Fire severity levels based on dNBR and CBI indices are reliable methodologies that allow for determining the severity and dynamics of changes in postfire regeneration levels in forested and nonforested areas.

**Keywords:** natural regeneration; Google Earth Engine; NBR; Chiquitano Forest; Abayoy

## 1. Introduction

Bolivia is among the countries with the highest wildfire activity recorded in South America in recent years [1,2]. In the last 20 years (2001–2020), wildfires in Bolivia directly affected an area of 23.6 million hectares, with the most extensive burnt areas being reported in 2004, 2005, 2010 and 2019 [3]. Annually, savannas are the most affected ecosystems by wildfires, followed by shrublands and forests [3]. Fire is mainly caused by human actions, principally due to pastures renewing the practice of slash and burn, known as *chaqueo* [4].

The Department of Santa Cruz, located in the eastern region of Bolivia, is one of the most affected by the wildfires, which mainly occur in the lowlands. In 2019, the worst occurrences of forest fires were recorded in Santa Cruz. As a result, 3.7 million hectares were affected, mainly in the Chiquitania region, of which 60% were forested areas [5]. These fires caused a series of ecological (e.g., impacts on soils, habitat losses, changes in species composition), economic (e.g., timber and wood products lost, agricultural production lost) and social (e.g., injured, air quality and smoke impacts) problems on the region, and directly affected the livelihoods of around 370 local communities [6]. In 2020, Santa Cruz recorded fires with 2.2 million hectares burned, of which 58% were forests [7]. These events reopened the debate about forest fires in the country, proposing different actions to recover

the affected areas. The regeneration of ecosystems impacted by the fires of 2019 and 2020 is a high priority and one of the greatest challenges in environmental terms, as much effort is needed to reconnect the impacted forest fragments and influence current land use change trends, especially with regards to biodiversity loss and the storage of important carbon reserves.

Faced with large-scale socio-environmental impacts generated after the events of 2019 in Chiquitania, the Autonomous Departmental Government of Santa Cruz, jointly with the National Government of Bolivia, developed the Plan for the Recovery of Fire-Affected Areas, which prioritized an area of 106,000 ha from the total affected territory to carry out assisted restoration [6]. However, the implementation of this plan involves obstacles due to the complexity of what it implies, together with its high costs. It is necessary to identify and evaluate alternatives that are feasible, operational, and effective. In this sense, one of the best proposals is to promote natural regeneration in areas impacted by fires, but it is initially necessary to measure how these processes occur.

Monitoring landscape changes has been one of the main uses of remote sensing over the past several decades [8]. The multiscale capability of remote-sensing equipment renders it particularly suitable for quantifying patterns of variation in space and time, detecting changes caused by both natural and anthropogenic disturbances [9], determining the extent of active fires, mapping fuel loads [10], and identifying areas with natural recovery [11]. To assess the recovery of vegetal communities and ecosystem effects through remote sensing, one needs to identify vegetation change trends [11–14], detecting postfire burn severity [15,16] and the vegetation spectral threshold [17–19], which can be obtained by analyzing the phenological patterns of vegetation over a time series [20]. However, few studies have examined postfire vegetation recovery in tropical ecosystems with Sentinel-2 time series analysis, especially for South America. Some dry tropical ecoregions of Bolivia (e.g., Chiquitano Dry Forest, Abayoy, Cerrado, and the Chaco) present challenges in determining regeneration levels due to annual and seasonal changes in vegetation phenological patterns [21].

The primary goal of this study is to determine dynamics of natural regeneration of ecosystems impacted by the wildfires of 2019 and 2020 at four areas in the Chiquitania region, based on (a) phenological patterns and (b) fire-severity-level analysis. Additionally, we selected sampling sites to apply the Composite Burn Index (CBI) methodology. These results help in determining which of the aforementioned remote-sensing techniques produce the most accurate products for assessing postfire ecosystem dynamics in the Chiquitano dry forest region, thereby contributing to a better understanding to support actions to enable natural regeneration.

## 2. Materials and Methods

### 2.1. Study Area

The Department of Santa Cruz comprises an area of 369,006 km$^2$, and is located between latitude 13°40′–20°20′ S and longitude 57°30′–64°40′ W, bordering the Department of Beni to the north, the Department of Cochabamba to the west, the Department of Chuquisaca, Paraguay to the south, and Brazil to the east. In this region, there is a very strong pattern of fire recurrence in anthropogenic areas [22], since its use in agriculture and cattle ranching is a common practice [23]. However, fire also impacted well-conserved spaces in indigenous territories [24] and protected areas between 2019 [25] and 2020 [7].

We conducted the study at four areas within the Department of Santa Cruz (Table 1, Figure 1): Alta Vista Tropical Dry Forest Study Center (Alta Vista), Copaibo de Concepción Natural and Cultural Heritage Municipal Reserve (Copaibo), Laguna Marfil Municipal Natural Area of Integrated Management (Laguna Marfil), and the indigenous Ñembi Guasu Protected Area of Conservation and Ecological Importance (Ñembi Guasu, Figure 2). The criteria for selecting these sites were based on the presence of burned areas between 2019 and 2020, favorable conditions for entry permits by local populations, levels of legal

protection, areas with productive agricultural or livestock activities, ease of access to sites impacted by fires, and ecosystem heterogeneity.

**Table 1.** Surface of burned areas in the four study sites in 2019 and 2020, Department of Santa Cruz, Bolivia.

| Study Area | Domain | Total Area (ha) | Burned Areas (ha) |
|---|---|---|---|
| Alta Vista | Private | 3425 | 525 ha (2019) |
| Copaibo | Municipal protected area | 347,037 | 36,628 ha (2019) 196,724 ha (2020) |
| Laguna Marfil | Municipal protected area | 70,916 | 22,175 ha (2019) 16,927 ha (2020) |
| Ñembi Guasu | Protected area of indigenous autonomy | 1,204,377 | 358,200 ha (2019) |

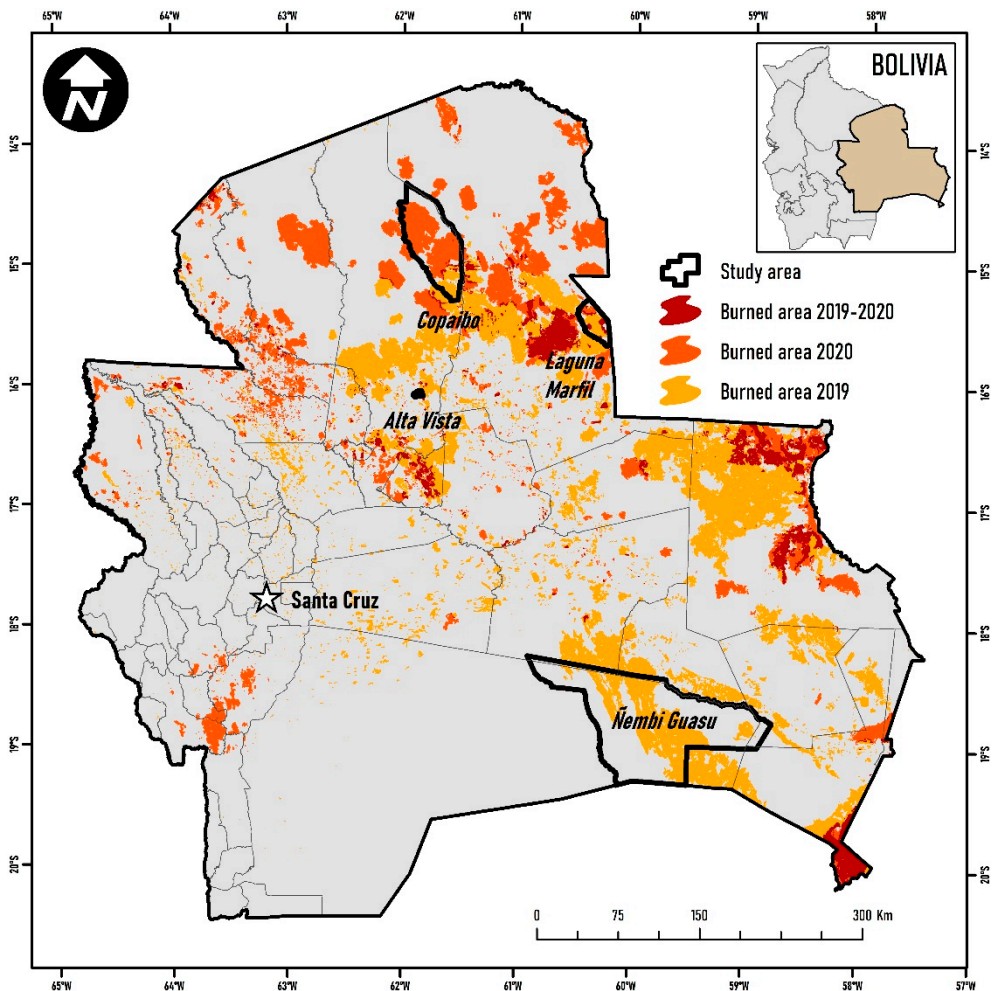

**Figure 1.** Burn scars in 2019–2020 and the four study sites in Department of Santa Cruz, Bolivia.

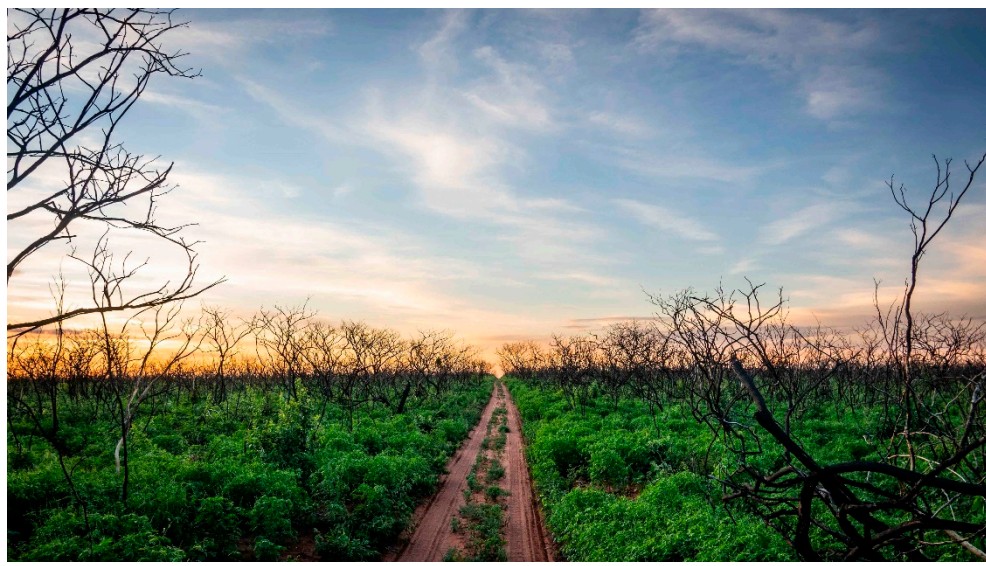

**Figure 2.** Abayoy, a scrubland formation impacted by fires in 2019 in the process of regeneration, indigenous Ñembi Guasu Protected Area of Conservation and Ecological Importance, Department of Santa Cruz (Photo: H. Azurduy).

### 2.2. Data Sources and Methods

For the different analyses, we used Sentinel-2 satellite data that provide images from 10 to 60 m spatial resolution with 13 spectral bands and five-day revisit frequency [26]. We generated fire scar maps for 2019 and 2020 from Sentinel-2 Level 1-C reflectance satellite imagery (TOA) with cloud cover less than 20%, which are publicly available on the USGS EarthExplorer portal (https://earthexplorer.usgs.gov, accessed on 10 March 2021. We performed different analyses in Google Earth Engine (GEE) with Sentinel-2 images to obtain the spectral indices of Normalized Difference Vegetation Index (NDVI [27]) and Normalized Burn Ratio (NBR [28]), using QA60 band to mask clouds. The NDVI shows values between –1 and 1, and is determined using the near-infrared (NIR) and red (RED) bands (Table 2), where values close to –1 indicate little photosynthetic activity and thereby little growth of or reduction in vegetation, while values close to 1 reflect the opposite. The NBR is obtained from the NIR and shortwave infrared (SWIR) bands (Table 2) that provides a better distinction between burned and unburned areas, as well as an optimal signal to obtain information on the variation of fire severity [28]. We used the land cover map (forest, nonforest, anthropogenic) elaborated by Maillard et al. [5] to intersect them with the 2019 and 2020 fire scar mapping, and thus determine the differences between burned and unburned natural areas. The obtained confidence level from the coverage classification was 85% [5].

**Table 2.** Spectral indices tested in the study, and their formulation and related spectral bands (band numbering refers to the Sentinel-2 sensor system).

| Index | Formulation | Bands |
|---|---|---|
| NDVI | (NIR − RED)/(NIR + RED) | NDVI (B4 − B8)/(B4 + B8) |
| NBR | (NIR − SWIR)/(NIR + SWIR) | (B8 − B12)/(B8 + B12) |
| dNBR | Prefire NBR − postfire NBR | |

### 2.3. Forest Fire Scars

We identified burned areas for 2019 and 2020 on the basis of the unsupervised classification of 56 Sentinel-2 scenes (Figure 1) for the July–December period using the Interactive Self-Organizing Data Analysis (ISODATA) algorithm [29] in ERDAS Image software to separate pixels into groups with similar spectral signatures. Previous studies that identified

forest fire scars in the Chiquitania region of Bolivia obtained accurate classification results using ISODATA [5,7]. Classification was performed with 100 maximal number of iterations, a convergence threshold of 100% (percentage of pixels whose class values should not change between iterations), and skip factors of 1. This resulted in a thematic classification of 25 spectral classes. A combination of image-specific RGB bands was also used for a visual review of the burned area detection and classification procedure. Obtained results were vectorized in ESRI ArcMap. To evaluate the level of uncertainty of the resulting classification, 250 field verification points were obtained between 2019 and 2021. In total, 66 sampling plots were obtained through the Composite Burn Index (CBI), 150 sampling points with ArcGisSurvey123 application, and 34 points with high-resolution images taken with UAV. This approach allowed for obtaining a confusion matrix [30] and three types of accuracy estimates: overall accuracy, user accuracy (commission error), and producer accuracy (omission error), including their 95% confidence intervals [30]. The overall accuracy obtained from the fire scars was 95.48% (Table 3).

**Table 3.** Confusion matrix and accuracy statistics.

| | Class | Reference Categories | | Total (*Wi* [a]) |
| --- | --- | --- | --- | --- |
| | | **Burned** | **Unburned** | |
| **Mapped categories** | Burned | 24.64 | 1.57 | 26.22 |
| | Unburned | 2.95 | 70.83 | 73.78 |
| | Total | 26.61 | 72.41 | 1 |
| | User's accuracy | 0.94 | 0.96 | |
| | Producer's accuracy | 0.89 | 0.98 | |
| | Overall accuracy | 95.48 | | |

[a] Class proportion with respect to total land area (2,203,379 ha).

### 2.4. Phenological Patterns of Burned and Unburned Areas

We evaluated the phenological patterns of vegetation in burned and unburned areas, identifying the thresholds of the NDVI and NBR indices. Both indices highlight changes in vegetation reflectivity when affected by fire. In GEE, we developed a script to perform probability sampling with random point spread, stratified by the cover type and proportion of burned and unburned areas for each study site. The sample number was calculated as proposed by Olofsson et al. [30], with a 90% confidence level for forests and 95% for nonforests. For the four study areas, a total of 1418 points were obtained, 393 for burned and 1025 for unburned areas. In GEE, we employed harmonic regression, a mathematical technique used to disaggregate a complex static signal into a series of individual sine or cosine waves, each characterized by a specific phase and amplitude [31]. We used the harmonic regression algorithms developed by Clinton [32] for Sentinel-2 collections to estimate the medians and harmonic fitted NDVI and NBR values for the selected land cover categories (Figure 3). The amplitude of harmonic regression is indicative of the magnitude of changes in spectral bands seasonally, which could be related to lapsing. To classify the phenological patterns of the mainland cover categories at different phases, we conducted time series analysis of the NDVI and NBR indices. We then identified the phenological patterns (monthly, seasonal, and annual) of burned and unburned areas, with the definition of average annual maximal and minimal thresholds for the period of 2016–2020, excluding outliers.

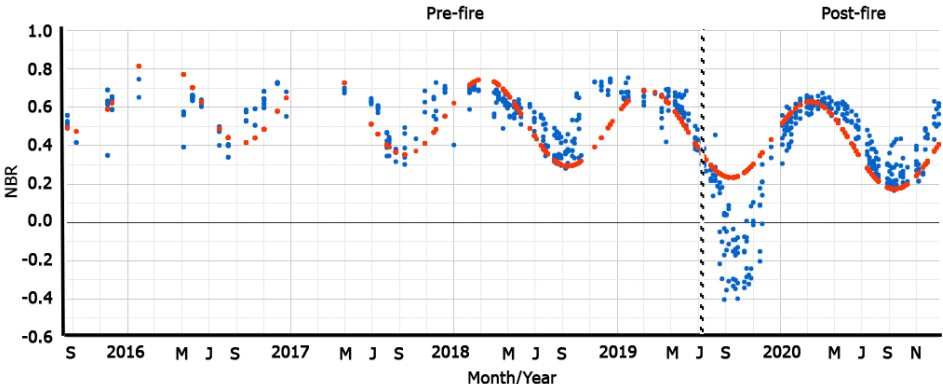

**Figure 3.** Example of original and fitted Sentinel-2 NBR temporal profiles of a deciduous forest burned of Ñembi Guasu generated by the harmonic model using GEE in the pre- and postfire periods (2016–2020). Red dots, fitted NBR values; blue dots, original NBR values.

### 2.5. Fire Severity Levels Based on dNBR

Fire severity is an important measure in determining postfire vegetation resilience and succession [33]. We quantified severity at the four study sites using the differential NBR or Delta NBR index (dNBR or ΔNBR, delta normalized burn ratio) obtained from the difference between pre- and postfire NBR (Table 2). Higher dNBR values indicate more severe damage, while areas with negative dNBR values may indicate new vegetation growth after a fire. On the basis of Key and Benson [28], we classified dNBR results into seven severity levels: enhanced regrowth/low (<−0.25), enhanced regrowth/high (−0.25 to −0.1), unburned (−0.1 to 0.1), low severity (0.1 to 0.27), moderate–low severity (0.27 to 0.44), moderate–high severity (0.44 to 0.66), and high severity (>0.66). Although the dNBR category thresholds proposed by Key and Benson [28] were developed for boreal ecosystems, we used these classes because they come from a globally standard widely used methodology to assess fire severity.

### 2.6. Fire Severity Based on Composite Burn Index (CBI)

The Composite Burn Index [28] (CBI) is based on a visual assessment of the quantity of consumed fuel, the degree of soil charring, and the degree of tree mortality [34], and it was applied to and adjusted in a wide variety of ecosystems [35]. Between 2019 and 2021, we surveyed burn severity in the four study sites (Table 4) using the CBI methodology. We evaluated fire affectation on the surface soil substrate; grasses, shrubs, and trees less than 1 m tall; shrubs and trees 1–5 m tall; intermediate trees 10–25 cm trunk diameter and 8–20 m tall; and large trees receiving direct sunlight [28]. We hierarchically aggregated attribute scores for the five strata from understory to canopy and then averaged the obtained values. We selected 66 sampling sites within the four study areas to apply the CBI (Table 4) aiming for heterogeneity in vegetation types within the identified burn scars. At each site, we installed 400 m$^2$ sampling plots at different levels of burn severity. In each plot, we described the site characteristics accompanied by four supporting photographic records, georeferenced to the four cardinal directions. The thresholds of CBI severity levels employed in this study were: (a) unburned = 0–0.25, (b) low = 0.25–1.25, (c) moderate = 1.25–2.25, and (d) high = 2.25–3 [36].

**Table 4.** Distribution of CBI plots for the evaluation of fire severity in the field in the four study areas in the Department of Santa Cruz.

| Study Area | Field Work | Evaluated Ecosystems | CBI Plots |
|---|---|---|---|
| Alta Vista | 7–8 October 2019, 27 October 2020 | Chiquitano Forest | 3 |
| Copaibo | 24–26 March 2021 | Chiquitano Forest transition to Amazon | 14 |
| Laguna Marfil | 23–26 October 2019, 13–16 February 2021 | Chiquitano Forest, Cerrado, Chiquitano Forest transition to Cerrado, Natural Savannah | 29 |
| Ñembi Guasu | 18–20 December 2019 | Chiquitano Forest transition to Chaco, Abayoy | 20 |
| Total | | | 66 |

*2.7. Statistical Analysis*

We used a two-way ANOVA test to evaluate significant differences in the values of the profiles fitted with the harmonic model of the Sentinel-2 NDVI and NBR indices for the five-year analytical period (2016–2020) in relation to burned versus unburned areas for the four study sites in the Department of Santa Cruz. The results of two-way ANOVA are shown in Table 5. To determine the relationship between the CBI and dNBR indices for 2019, Spearman's correlation analysis was performed ($p < 0.05$). We used RStudio [37] for all statistical analyses.

**Table 5.** Two-way ANOVA results on difference between NDVI and NBR spectral indices in the five years of analysis (2016–2020) in burned versus unburned areas in the four study areas of the Department of Santa Cruz. Degrees of freedom (*Df*), mean sum of squares (*MS*), F-statistic (*F*), and *p*-value (*p*) are the test parameters.

| Study Area/ Ecosystem | Index | Factor | *Df* | *MS* | *F* | *p* | |
|---|---|---|---|---|---|---|---|
| Alta Vista/forest | NDVI | Years | 4 | 0.22 | 17.75 | 0.000 | *** |
| | | Burned vs. unburned | 1 | 0.01 | 0.96 | 0.328 | |
| | | Years x burned vs. unburned | 4 | 0.02 | 1.47 | 0.208 | |
| Alta Vista/forest | NBR | Years | 4 | 0.05 | 10.61 | 0.000 | *** |
| | | Burned vs. unburned | 1 | 0.03 | 6.65 | 0.010 | * |
| | | Years x burned vs. unburned | 4 | 0.02 | 3.84 | 0.004 | ** |
| Copaibo/forest | NDVI | Years | 4 | 2.27 | 84.25 | 0.000 | *** |
| | | Burned vs. unburned | 1 | 0.06 | 2.26 | 0.133 | |
| | | Years x burned vs. unburned | 4 | 0.24 | 8.73 | 0.000 | *** |
| Copaibo/forest | NBR | Years | 4 | 4.11 | 127.53 | 0.000 | *** |
| | | Burned vs. unburned | 1 | 2.79 | 86.72 | 0.000 | *** |
| | | Years x burned vs. unburned | 4 | 0.75 | 23.28 | 0.000 | *** |
| Laguna Marfil/forest | NDVI | Years | 4 | 0.10 | 5.11 | 0.000 | *** |
| | | Burned vs. unburned | 1 | 0.11 | 5.70 | 0.017 | * |
| | | Years x burned vs. unburned | 4 | 0.02 | 0.97 | 0.424 | |
| Laguna Marfil/forest | NBR | Years | 4 | 0.11 | 6.73 | 0.000 | *** |
| | | Burned vs. unburned | 1 | 0.14 | 8.65 | 0.003 | ** |
| | | Years x burned vs. unburned | 4 | 0.04 | 2.47 | 0.043 | * |
| Laguna Marfil/nonforest | NDVI | Years | 4 | 0.04 | 2.68 | 0.031 | * |
| | | Burned vs. unburned | 1 | 0.51 | 35.03 | 0.000 | *** |
| | | Years x burned vs. unburned | 4 | 0.01 | 0.92 | 0.454 | |

**Table 5.** *Cont.*

| Study Area/Ecosystem | Index | Factor | Df | MS | F | p | |
|---|---|---|---|---|---|---|---|
| Laguna Marfil/nonforest | NBR | Years | 4 | 0.10 | 5.39 | 0.00 | *** |
| | | Burned vs. unburned | 1 | 1.02 | 56.91 | 0.00 | *** |
| | | Years x burned vs. unburned | 4 | 0.07 | 4.04 | 0.00 | ** |
| Ñembi Guasu/forest | NDVI | Years | 4 | 1.16 | 43.18 | 0.000 | *** |
| | | Burned vs. unburned | 1 | 0.20 | 7.50 | 0.006 | ** |
| | | Years x burned vs. unburned | 4 | 0.17 | 6.45 | 0.000 | *** |
| Ñembi Guasu/forest | NBR | Years | 4 | 1.96 | 65.33 | 0.000 | *** |
| | | Burned vs. unburned | 1 | 3.25 | 108.03 | 0.000 | *** |
| | | Years x burned vs. unburned | 4 | 0.47 | 15.51 | 0.000 | *** |
| Ñembi Guasu/nonforest | NDVI | Years | 4 | 1.13 | 42.28 | 0.000 | *** |
| | | Burned vs. unburned | 1 | 0.02 | 0.71 | 0.398 | |
| | | Years x burned vs. unburned | 4 | 0.08 | 2.90 | 0.021 | * |
| Ñembi Guasu/nonforest | NBR | Years | 4 | 2.18 | 65.87 | 0.000 | *** |
| | | Burned vs. unburned | 1 | 0.23 | 6.98 | 0.008 | ** |
| | | Years x burned vs. unburned | 4 | 0.30 | 8.93 | 0.000 | *** |

*, significant difference at $\alpha = 0.05$; **, significant difference at $\alpha = 0.01$; ***, significant difference at $\alpha = 0.001$.

## 3. Results

### 3.1. Phenological Patterns of Burned and Unburned Areas

Analysis of the phenological patterns based on the fitted values of the harmonic model for Sentinel-2 NDVI and NBR showed marked seasonal (rainy season and dry season) and interannual variability (Figure 4) in the forested and nonforested areas of three of the four sites. The highest photosynthetic activity was recorded in the months of January–May and November–December, while June–October showed low chlorophyll absorption (Figure S1). In the two-way ANOVA test, significant differences were evident, mainly for NBR, when comparing years, and burned and unburned areas (Table 5). These results indicate that NBR is the most reliable index for interannual comparisons and determining changes in the phenological pattern (Figure 4). The main changes in NBR were between the minimal threshold levels of the prefire period (2016–2018) and the postfire values (2019–2020).

At AltaVista, interannual comparisons indicate a change in the minimal threshold of NBR in the forest that was impacted by fire, where the prefire stage recorded an average value of 0.31, which decreased to 0.16 in 2019 and reached 0.32 in 2020 (Figure 4), demonstrating a regeneration process one year after the fire. At Copaibo, the average minimal NBR threshold of the burned forest in the prefire stage was 0.61, which increased to 0.54 in 2019 and 0.34 in 2020. According to our analysis, no regeneration processes were detected in Copaibo due to the continuous fires in 2020 (Figure 4). At Laguna Marfil, interannual comparisons show some changes in the phenological patterns of the forest, with an increase in the minimal threshold from 0.34 in 2016–2018 to 0.22 in 2020 (Figure 4). It is evident that there is a regeneration process in this forest. However, in the nonforested areas of Marfil, the average values during the pre- and postfire periods were 0.14 and –0.15, respectively, indicating no regeneration (Figure 4). This was due to registered fires in two consecutive years (2019–2020). In Ñembi Guasu, the burned forest shows changes in the minimal NBR threshold, from a value of 0.31 in the prefire period to −0.43 (2019) and 0.17 (2020) in the postfire period (Figure 4). In the nonforested areas (Abayoy) of Ñembi Guasu that had been impacted by fire, the average value of the minimal NBR threshold was 0.25 (prefire), which increased after the fire event to −0.36 in 2019 and 0.17 in 2020. A regeneration process is evident in both types of Ñembi Guasu ecosystems.

### 3.2. Fire Severity Levels Based on dNBR

Values obtained in the dNBR for fire scars from the four study areas indicate a natural regeneration scenario between 2019 and 2020, even though some severity levels were not reduced because some sites burned in 2019 and 2020 (Figures 5–8, Table S1). At Alta Vista, the fire categories with the highest proportion in 2019 were low severity (46%) and unburned (27%); however, by 2020, there was an increase in dNBR for the lower-severity categories, such as enhanced regrowth/high with 64% and enhanced regrowth/low with 34% (Figure 5, Table S1). At Copaibo, fire severity levels in 2019 were mainly concentrated in the categories of moderate–low severity with 37%, low severity with 27%, and moderate–high severity with 24%. Although a reduction in severity levels was expected, fires in 2020 caused these two categories to continue to present the highest percentages with 21%, 24% and 22%, respectively (Figure 6, Table S1). At Laguna Marfil, the dNBR levels for 2019 and 2020 indicate a reduction in severity levels in moderate–low severity from 51% to 11%, and low severity from 25% to 17%; in 2020, the unburned and enhanced regrowth/high classes increased to 24% and 26%, respectively (Figure 7, Table S1). At Ñembi Guasu, the severity levels of fires in 2019 based on dNBR were mainly concentrated in the moderate–high-severity category with 32%, and in the categories of moderate–low severity with 20% and moderate–high severity with 18%. By 2020, the low-severity (40%) and unburned (25%) categories increased considerably, demonstrating the recovery of vegetation in this protected area (Figure 8, Table S1).

### 3.3. Fire Severity Based on CBI

Of the 66 CBI plots that we installed in the four study areas between 2019 and 2021, the 16 that we remeasured showed a natural regeneration process (Table S2). At Alta Vista, the three CBI plots in the Chiquitano Forest demonstrated moderate severity levels in 2019 (CBI = 1.4–2), and subsequently resulted in reduced severities of low and unburned categories in 2020 (CBI = 0.1–0.8) (Table S2). At Copaibo, 3 of the 14 severity plots that we measured in 2021 were not affected by the 2019 or 2020 fires (Table S2), so the CBI values were very low (<0.4); the remaining 11 showed severity levels in the unburned-to-moderate categories (CBI = 0.4–2.1).

At Laguna Marfil, we installed 10 CBI plots in the Chiquitano Forest between 2019 and 2021, and observed that, in three plots, levels changed from low/moderate (CBI = 0.1–1.5) to low (CBI = 0.0–0.5) (Table S2). For this forest type, we measured seven new plots in 2021, which showed greater variability (CBI = 0.0–2.8) (Table S2). In the Chiquitano Forest transition to Cerrado of Marfil (Table S2), we implemented three CBI plots in 2021, in which we identified low severity levels (CBI = 0.6–0.9). In the Cerrado vegetation plots of Marfil, five of the seven field plots showed changes from low/high severity category levels (CBI = 0.4–2.5) in 2019, reducing to moderate/unburned levels (CBI = 0.0–2.0) in 2021. In the natural savannah of Marfil, all five CBI plots showed low/moderate severity levels (CBI = 0.3–1.8), reducing to moderate-to-unburned levels (CBI = 0.0–1.4) in 2021 (Table S2).

At Ñembi Guasu, the 20 established CBI plots were located in areas impacted by fires in 2019 (Table S2). For the scrubland formation known as Abayoy, the field evaluations that we had conducted months after the 2019 fire series showed high severity levels (CBI = 2.3–2.7). In the Chiquitano Forest transition to Chaco of Ñembi Guasu (Table S2), fire severities also showed high levels (CBI = 2.4–2.7). Of the total sampling in the four study areas, both ecosystems presented the highest levels of severity.

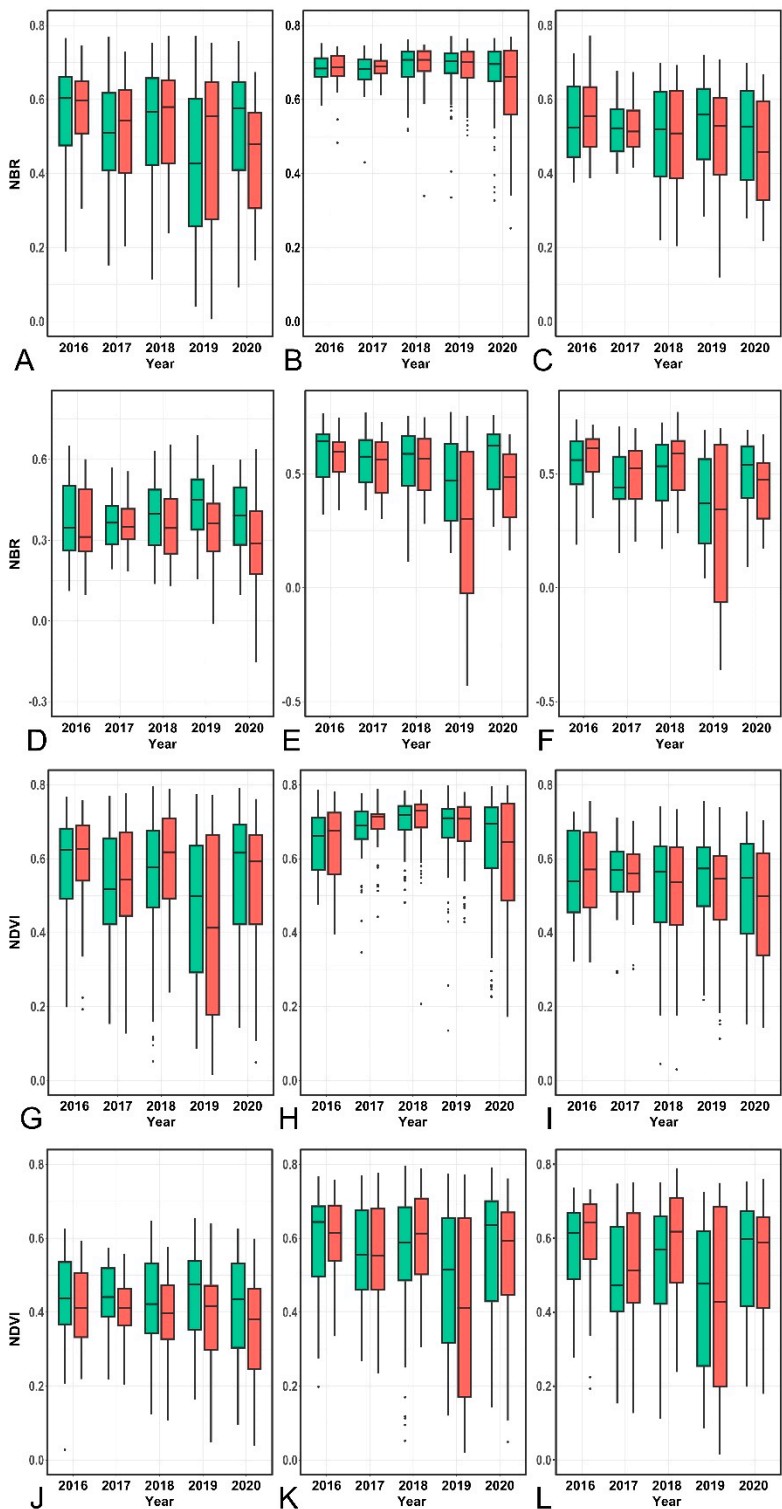

**Figure 4.** Boxplots of unburned (green) and burned (red) areas. Average annual maximal and minimal thresholds of spectral indices for NBR at Alta Vista ((**A**), forest), Copaibo ((**B**), forest), Laguna Marfil ((**C**), forest; (**D**), nonforest) and Ñembi Guasu ((**E**), forest; (**F**), nonforest). Average annual maximal and minimal thresholds of spectral indices for NDVI at Alta Vista ((**G**), forest), Copaibo ((**H**), forest), Laguna Marfil ((**I**), forest; (**J**), nonforest) and Ñembi Guasu ((**K**), forest; (**L**), nonforest). Box plot elements: box = values of 25th and 75th percentiles; horizontal line = median; whiskers = values of 5th and 95th percentiles.

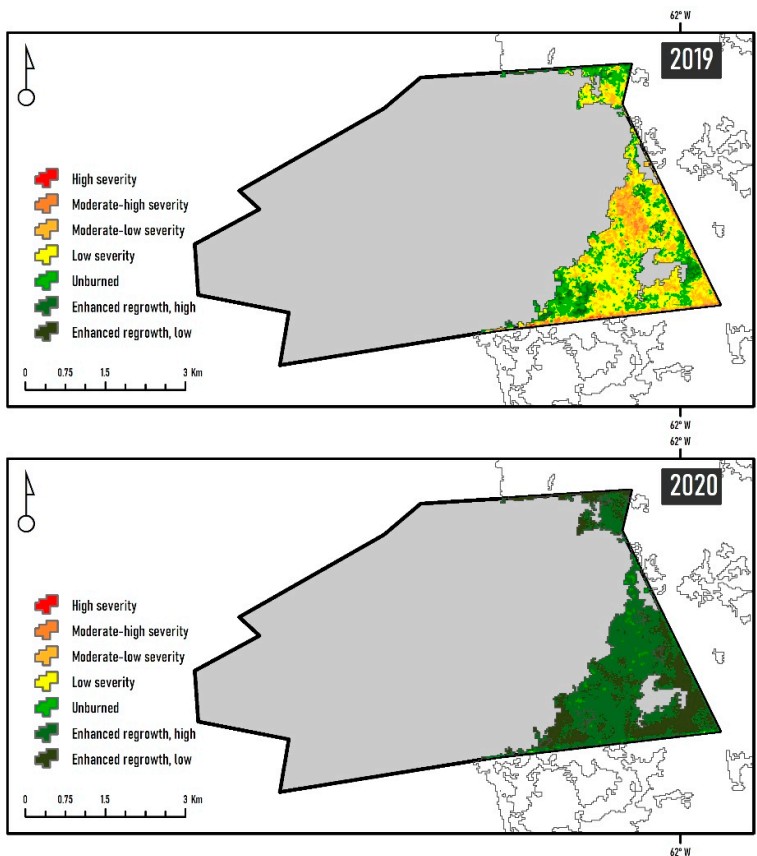

**Figure 5.** Fire severity levels based on dNBR at Alta Vista between 2019 and 2020.

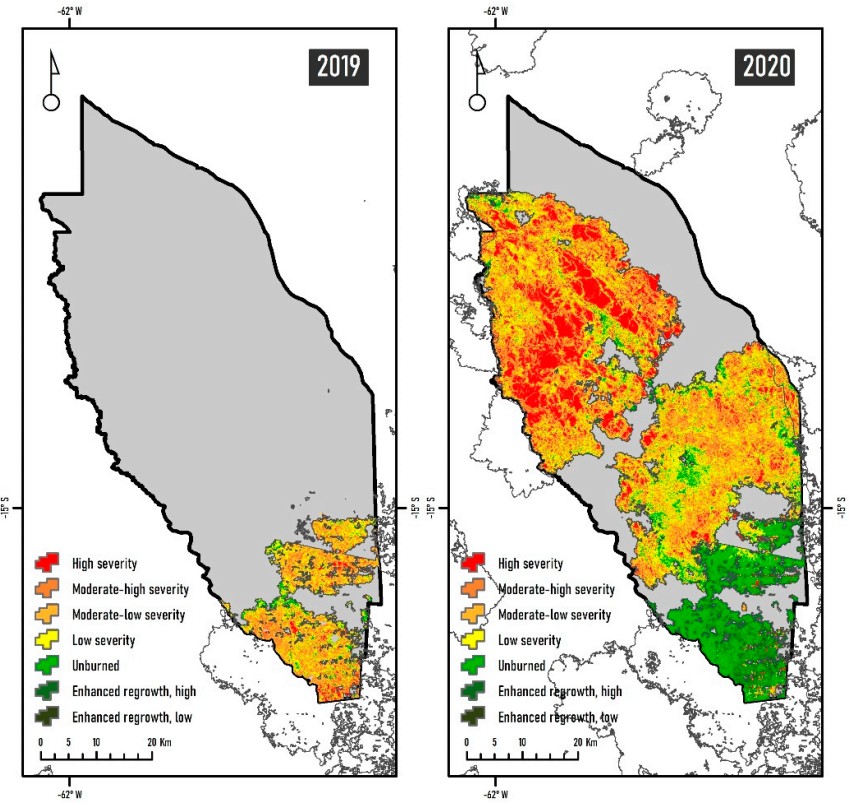

**Figure 6.** Fire severity levels based on dNBR at Copaibo between 2019 and 2020.

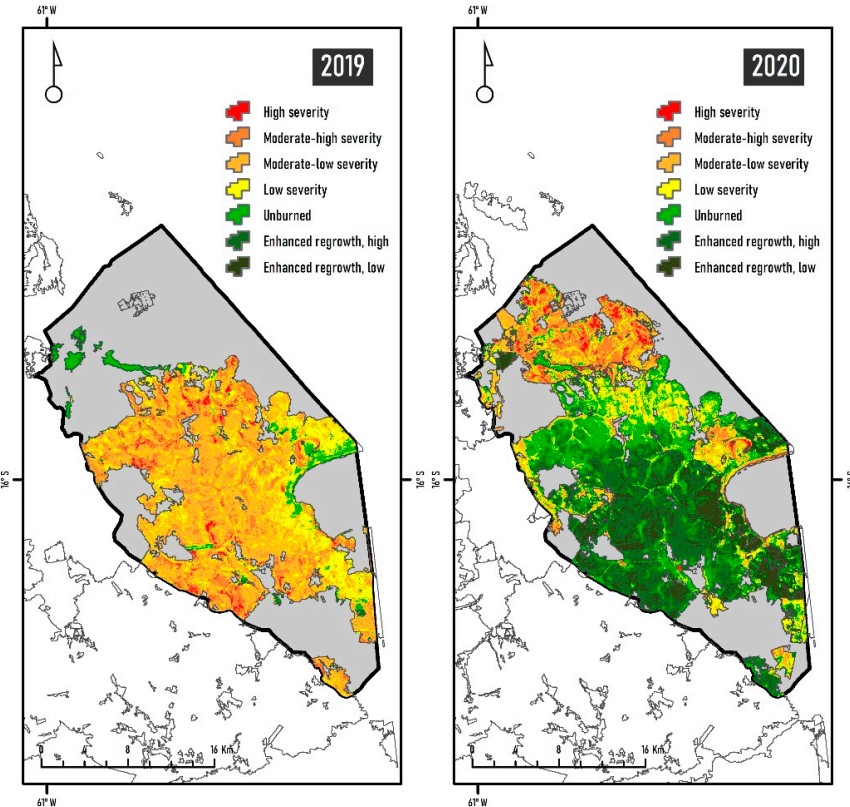

**Figure 7.** Fire severity levels based on dNBR at Laguna Marfil between 2019 and 2020.

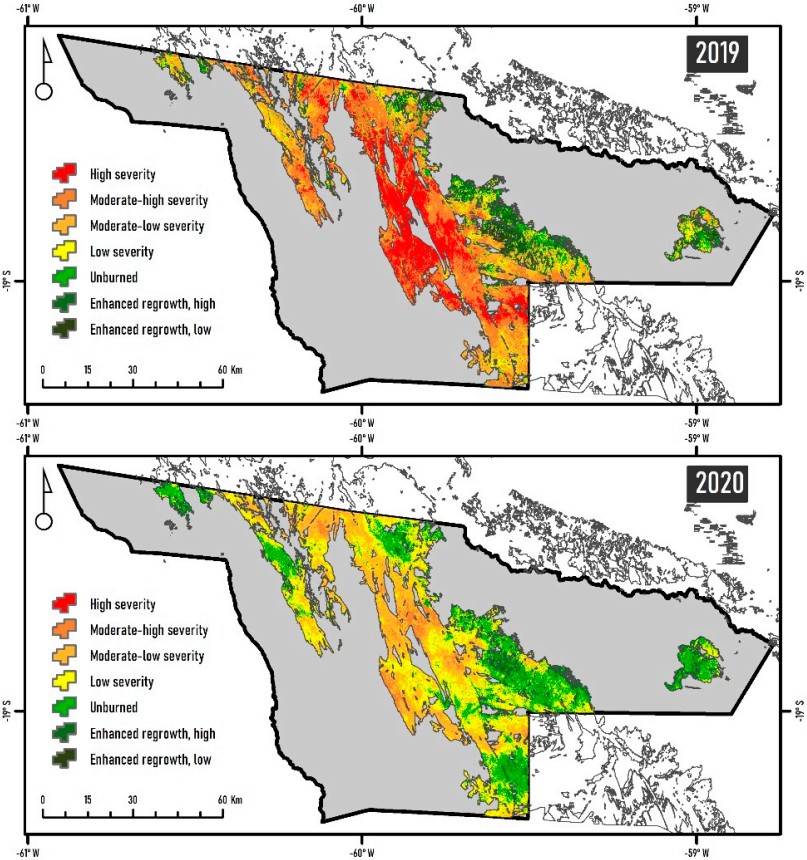

**Figure 8.** Fire severity levels based on dNBR at Ñembi Guasu between 2019 and 2020.

Relationship analyses between the dNBR and CBI indices for burned forested and nonforested areas, show a significant and high correlation for 2019 ($R = 0.82$, $p < 0.001$) (Figure 9). This indicates that the results obtained with the dNBR can be considered reliable.

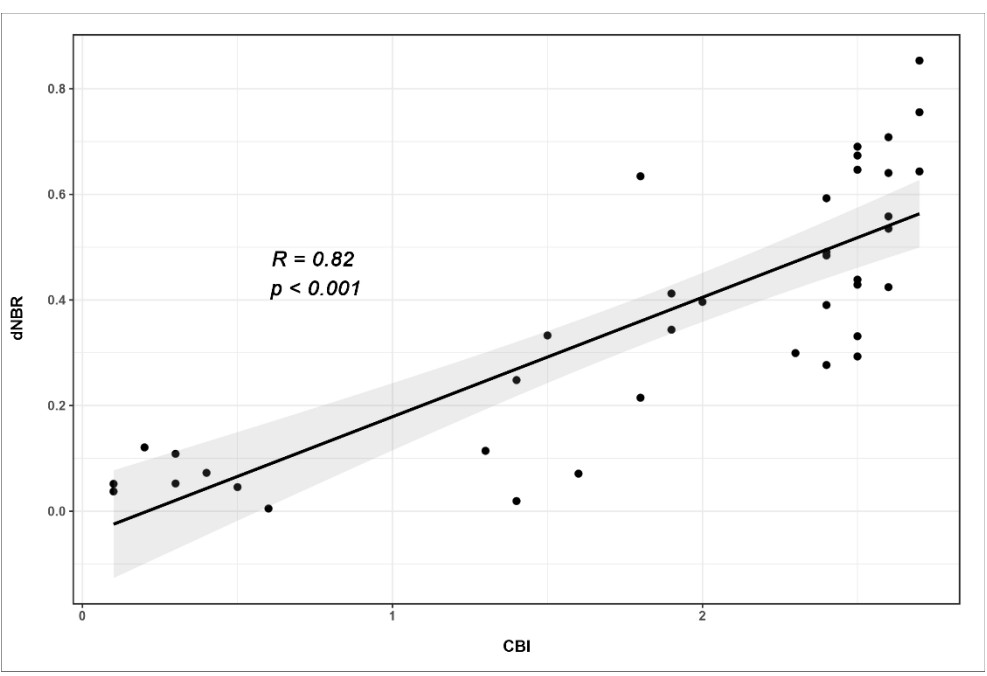

**Figure 9.** Correlation between CBI values and dNBR with Pearson correlation coefficients for 2019.

## 4. Discussion

Wildfires reduce the reflectance of visible-to-NIR wavelengths due to the loss of photosynthetic vegetation, and increase the reflectance of SWIR wavelengths due to increased char and ash [38,39]. The postfire vegetation regeneration processes had the opposite effect on spectral reflectance. In this sense, we calculated and evaluated the NDVI and NBR indices, and compared the results for burned and unburned areas, from forested and non-forested ecosystems, allowing for us to determine the index that offers the best capabilities to reflect postfire regeneration in the ecosystems of the Chiquitania region. The forested cover of the four study areas had a much higher NDVI and NBR than those of nonforested areas, which decrease during the dry season. However, annual differences were recorded for the period 2016–2020, between the averages of the maximal and minimal thresholds in unburned areas. These differences are related to a response of the vegetation to annual seasonal characteristics, such as temperature, precipitation, and humidity. In months with low photosynthetic activity, the vegetation enters a dormancy process to save energy during the dry season, as had been recorded in other deciduous forests in Bolivia [21]. It is important to consider these results to avoid errors in the interpretation of spectral thresholds in the comparison of burned and unburned areas. In addition, we must consider meteorological drought events that were recorded in recent years on a continental scale [40], which were also perceived and recorded in Bolivia [5]. Our results show that the NBR showed a better spectral response, as it allowed for us to identify significant differences between years, and between burned and unburned areas because the red/NIR-based vegetation indices are mainly related to the density of green leaves in a given area, and are thereby a good indicator of plant cover and vitality [41]. The SWIR-based vegetation indices are sensitive to the detection of dead or nonphotosynthetic vegetation in a postfire environment [42]. In other regions of the world, recent studies showed that NDVI is effective in monitoring postfire vegetation succession [43,44]; however, our results indicate that NBR is the most reliable index for interannual comparisons and determining changes in the phenological pattern in this region of Bolivia, which allows for the detection of postfire regeneration.

Fire severity levels based on dNBR and CBI indices are reliable methodologies that allow for determining the severity and dynamics of changes in postfire regeneration levels in forested and nonforested areas, as well as in their transitional stages. In the case of the dNBR, it is necessary to consider some important aspects, such as comparisons between the same stations. In Chiquitania, fires generally occur during the dry season [45,46]; after the first rains, there is a higher concentration of moisture in the soil, and vegetation resprouts develop, which could be interpreted as an apparent complete regeneration. In this study, we compared differences in the severity levels of dNBR after 1–2 years, considering seasonality. The dNBR index decreases over time as an effect of the disappearance of fire disturbance. In the case of CBI, the size of the plots and the practicality in data collection render it a replicable methodology to any site in the Chiquitania. Furthermore, we found high and significant correlation between the CBI and dNBR indices for 2019. The majority of studies showed medium-to-high correlation between CBI/GeoCBI severity levels and NBR–dNBR values [47–51]. However, all these correlation analyses between the two indices were performed in boreal ecosystems, and information for tropical ecosystems is almost nonexistent, so our research contributes to determining the reliability of using severity analysis based on remote sensing. In addition, our research was based on the classification proposed by Key and Benson [28], who defined the thresholds on the basis of research in boreal ecosystems, so that future research should be directed to define the categories of forest fire severity in tropical ecosystems.

Differences in fire severity levels in the same area are not fully understood. Some aspects that should be evaluated in future studies to assess these differences should consider the type of fire (ground, surface or crown fires), meteorological conditions, the moisture concentration of the biomass, and the type of forest fuel [52]. In certain forested regions of the Chiquitania, some fire-adapted species (e.g., bamboo *Guadua paniculata*), generates high levels of fine biomass, and increases fire intensity and forest flammability [53,54]. In the nonforested areas adapted to fire in the Laguna Marfil and Ñembi Guasu (e.g., Cerrado, Abayoy, natural savannah), the fires consumed the greatest amount of fine and medium biomass; in the forested areas of the four study areas, they were surface fires, but in certain areas of the forest, the appearance of crown fires was evident, demonstrating higher levels of severity dNBR and CBI, but without reaching the most extreme values recorded. In Ñembi Guasu, the protected area where a series of magnitude fire events with a high magnitude were recorded [55,56], we determined that severity levels were the most extreme for 2019, due to the high intensity of wildfires [3].

A fundamental topic in spatial forest ecology that is related to fire is mapping and modeling the complexity of postburn forest patterns and their changes over time [20]. The results yielded in our research show that the forested and nonforested ecosystems of the Chiquitania impacted by fires, are in the process of natural regeneration, whereas the only human interventions were the avoidance of degradation processes. Therefore, we recommend a more extensive protection and monitoring of the affected areas to ensure a successful natural regeneration and to avoid a recurrence of fires, which would reduce a successful regeneration of the areas impacted by the 2019–2020 fires. Although the results of the present work do not yet allow f or us to determine the time of complete recovery of the forests in terms of composition and structure because the growth of the tree canopy occurs much more slowly than the recovery in vegetation indices does. For this reason, it is necessary to continue monitoring the sites on the basis of remote-sensing and long-term field evaluations to better understand the dynamics, trends of change over time, and the ability of the different ecosystems of the Chiquitania to recover.

## 5. Conclusions

After the 2019 and 2020 fires, a process of natural regeneration of the ecosystems is evident among the four study sites. However, some regions are in a state of degradation, mainly in areas where the severity of the fires was high (Ñembi Guasu) and where these fire events were repeated in two consecutive years (Laguna Marfil and Copaibo). These results

should be complemented with further studies, since it is necessary to continue developing long-term monitoring mechanisms that allow for us to deepen our understanding of fire ecology and natural regeneration processes in these sites. Our results demonstrate the importance of using monitoring technologies including remote sensing assessments in combination with targeted and cost-effective field work to obtain more accurate information on the responses of impacted ecosystems, both in terms of their ecological characteristics and the degree of intensity and repetitiveness of wildfires.

We recommend precautionary restoration action for areas affected by fires, and the establishment of safeguards and guarantees for the development of natural regeneration processes, as an action properly delimited, planned, and implemented according to the respective legal, technical, and administrative guidelines. Lastly, there is a clear need to establish a scientific and technical research agenda related to methodologies and analyses to evaluate scenarios in the framework of restoration.

**Supplementary Materials:** The following supporting information can be downloaded at: https://www.mdpi.com/article/10.3390/fire5030070/s1, Figure S1: Monthly phenological patterns of burned and unburned areas based on Sentinel-2 NDVI for 2019 and 2020 for the four study areas. Table S1: Fire severity levels in hectares and percentages (in parentheses) based on Sentinel-2 dNBR for 2019 and 2020 for the four study areas; Table S2: Fire severity levels based on CBI plots in the four study areas of the department of Santa Cruz.

**Author Contributions:** Conceptualization, O.M., R.V.-A. and H.A.; methodology, O.M., M.F.-V., G.M., R.C. and M.B.; software, O.M., M.F.-V., G.M. and R.C.; validation, O.M., M.F.-V., G.M., R.C., H.A., R.F. and S.A.; formal analysis, O.M., M.F.-V., G.M., R.C. and M.B.; investigation, O.M., M.F.-V. and G.M.; resources, O.M., M.F.-V. and G.M.; data curation, O.M., M.F.-V. and G.M.; writing—original draft preparation, O.M., M.F.-V. and G.M.; writing—review and editing, O.M., H.A. and N.M.; visualization, O.M., M.F.-V. and G.M.; supervision, H.A. and R.V.-A.; funding acquisition, R.V.-A. All authors have read and agreed to the published version of the manuscript.

**Funding:** This study was developed as part of a series of investigations by the FCBC Chiquitano Forest Observatory in the framework of the ECCOS project, financed by the European Union, and the Knowledge Bases for Restoration project, financed by the Government of Canada.

**Institutional Review Board Statement:** Not applicable.

**Informed Consent Statement:** Not applicable.

**Data Availability Statement:** Data are contained within the article.

**Acknowledgments:** The authors are grateful to Edgar Viveros (FCBC), Tito Arana (FCBC), Oscar Cambara. Pedro Cambara, Celestino Cambara (Laguna Marfil), Ruben Ortiz (Gobierno Autónomo Indígena Originario Campesino de Charagua), Florencio Mendoza, Donald Viera y Osvaldo Cunay (Alta Vista), Henry Condori (Campo Esperanza, Copaibo) for their assistance during the field campaigns. We are also grateful to Carla Pinto Herrera and Eric Fox, for their valuable help. We thank four anonymous reviewers for their insightful comments and suggestions that greatly helped in improving our manuscript.

**Conflicts of Interest:** The authors declare no conflict of interest. The funders had no role in the study design; in the collection, analysis, or interpretation of data; in the writing of the manuscript; or in the decision to publish the results.

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
