# Peer review of "Phenology Patterns and Postfire Vegetation Regeneration in the Chiquitania Region of Bolivia Using Sentinel-2"

_fire, doi:10.3390/fire5030070_

Round 1

Reviewer 1 Report

This is the second time I am reviewing the manuscript, entitled in its revised form “Phenology Patterns and Post-Fire Vegetation Regeneration in the Chiquitania Region of Bolivia Using Sentinel-2”. In my first review I stated that this is an interesting case study worth to be published, despite the fact that the methods employed are not novel. However, I had some major concerns regarding the structure of the manuscript and its consistency. This revised version, retains the main finding, which is that these ecosystems promptly respond to wildfire impacts by natural regeneration. Based on this result the authors make a claim towards promoting natural regeneration, as opposed to artificial planting, and ensuring the protection of the affected areas from further degradation.

The structure of the manuscript has improved a lot in this revised version, and although, in my personal view, it could be further improved, it is now in a state that it can be published. It reads well throughout and apart from a few minor linguistic points it is well written. It still reports a high number of numbers in the results but it has improved since its last version. Also in the results, I would add the Apendix table 1, where the ANOVA results are reported, in the main text. I was surprised to see non-significant differences in the NDVI values between burned and unburned sites in some cases, especially since recent studies (e.g. Fire 2021, 4(4), 92; https://doi.org/10.3390/fire4040092; Fire 2021, 4(4), 64; https://doi.org/10.3390/fire4040064 ;  Remote Sensing of Environment,  2016, 183, 53-64, https://doi.org/10.1016/j.rse.2016.05.018 ) demonstrates that demonstrates that NDVI is effective in monitoring post fire vegetation succession. However, since this is the result it is what it is.

Overall, I believe the study is worth to be published and the authors may consider the few points I mentioned above regarding the results section.

Author Response

Thanks for your positive assessment of the manuscript and the constructive comments. These comments have contributed to improving the readability and clarity of the manuscript.

Attached is a point-by-point response to the reviewer's comments. 

Reviewer 2 Report

This manuscript presents a comprehensive analysis of post-fire vegetation regeneration. The data and methods used are described in great detail. My only reservation is regarding the Figures' and Tables' captions, which are needed more information. For example, consider adding what B4, B8, etc. mean in Table 2.

I spotted two minor typos:
line 26 - check the word methodology
line 79 - delete repeated in

I recommend the manuscript for prompt publication.

Author Response

(The authors gave the same response as above.)

Reviewer 3 Report

The article provides the RS based monitoring of the forest ecosystem. The findings are interesting but the study is not that innovative. There are several serious issues throughout the article in addition to significance problem that are raised below:

  1. Title needs modification, it does not reflect what the paper does. needs monitoring term after the regeneration and needs satellite images after Sentinel 2.
  2. Abstract does not reflect the used data and methodological approach. Also, it needs numerical results related to accuracy.
  3. Keywords should also reflect the dataset and methods
  4. Introduction is very limited and needs further literature review of RS based attempts and different approaches  in a categorized way. At the end, it requires a statement providing info about what is the significance of this study and which gap(s) filled when compared to previous researches.
  5. FÄ°gure 2 may not be needed in current form. Maybe it will be beneficial when more region samples with smaller figures are combined together.
  6. Title 2.2 title should be revised including a term of "data".
  7. ISODATA unsup. classification is a very old method, thus its performance (accuracy) is questionable when compared to the current state-of-the-art methods. Moreover, this analysis should be verified by an accuracy assessment procedure.
  8. 8. Excel and Arcgis data analysis steps are not required, there should be more focus on algorithms  / methods instead of software.
  9. Authors performed a dNBR based thresholding, but they used the thresholds coming from an older paper (Key & Benson). However, such thresholds are study and region specific so they should be validated and tuned according to current study. This part needs further process.
  10. Figure 4 has some cartographic problems. Legend in the right figure overlays the analysis region, text and scales are not readable, sub numbering is required (a,b,c, etc.), the symbol for CBI plots are not good and big in size.
  11. Figure 6 has similar problems.
  12. Table A1, test parameters should be explained in the text in detail.
  13. I strongly recommend a workflow diagram as a figure.

Author Response

(The authors gave the same response as above.)

Reviewer 4 Report

Phenology Patterns and Post-Fire Vegetation Regeneration in the Chiquitania Region of Bolivia Using Sentinel-2

 Maillard et al. (2022) have studied fire recovery in Bolivia after the 2019 and 2020 fires using remote sensing and field data. The thematic involved in this article is within the scope of Fire and would be interesting to the readers of the journal.  

In my opinion, this manuscript needs several clarifications before being suitable for publication. The most critical part of the manuscript are the methods, where several important steps are not properly described. The CBI method and results are not easily understandable, for example. Moreover, how did you validate the burned scars’ automatic mapping? Please see the comments below:

Lines 37-38: Are deforestation fires in the Bolivian portion of Amazonia significant?

Line 39: Where is this department located? It is not familiar to those outside Bolivia.

Lines 40-41: Were these fires associated with the 2019 Amazonia burning crisis? And the 2020 Pantanal drought?

Lines 42-43: Please give some examples of these problems.

Lines 52-55: A map showing this information would be great.

Line 58: Could not understand. Are you proposing an entire new plan or additional actions to the current pan?

Line 68: “trends in change” is awkward. Please rephrase it.

Line 76: Please briefly explain this method. What CBI measures?

Line 79: Please exclude “in”.

Line 89: Could not understand the point of this figure. Perhaps it could be deleted?

Line 92: Please better elaborate the importance of fires in your study area, major sources of ignition and purposes. I assume that fires are forbidden in protected areas, for example.

Line 98: Please add coordinates to this map. The red color represents burned in 2019 and 2020?

Lines 109-111: How many images were processed?

Lines 111-114: Awkward sentence. Please rephrase it.

Line 117: Please elaborate on the effect of soil in NDVI in here.

Line 125: Please use the spectral region (e.g., SWIR) instead of the band number in here. This changes according to the sensor. Moreover, add the reference of each index in the table.

Lines 128-136: You must describe the method adopted to map burned area in here. For example, what is this threshold representing? This is a critical part of the manuscript.

Lines 128-136: How did you validate the burned scars’ automatic mapping?

Lines 143-145: Please elaborate on the proposition made by Olofsson et al. in this sentence.

Lines 156-158: How the thresholds were defined? Not clear so far.

Lines 166-169: How did you define the burning date?

Line 178: Please add “-” before 0.25.

Lines 186-188: How did you define these strata?

Lines 210-211: Rainy season and dry season?

Lines 218-244: Not sure what you meant by thresholds. Please explain. Are the values in Figure 5 the estimated values?

Line2 275-276: What did you meant by “remediated”?

Lines 317-319: Was your study area impacted by the Pantanal drought in 2019 and 2020?

Lines 343-345: What about the meteorological conditions?

Lines 413-416: Please explain the test parameters in this caption.

Author Response

(The authors gave the same response as above.)

Round 2

Reviewer 4 Report

I appreciate the authors’ effort on addressing most of my comments. However, some of them still deserve a proper answer and modification on the manuscript. These are the ones related to the burned areas’ mapping. It is not clear so far if you have performed an entirely new classification or used one from a previous study. It seems like a new one from section 2.3 but you cite and use confidence levels from a previous study. This is problematic since this validation is not applicable to a new estimate of burned area. It is the core of your study and deserves to be treated so. Please include proper explanations on the method, validation, and thresholds for aping burned areas. I strongly agree with Reviewer 3 on this. Moreover, although you believe it is not necessary to add coordinates to the map in Figure 1 they are. So, one can properly identify your study area. Please add them.  

Author Response

We thank once again reviewer #4 for his valuable comments and suggestions.

In this version we are including more information on method 2.3 and we have also included information on the accuracy estimates assessment. We have included a map with coordinate grid.

Thank you very much

Round 3

Reviewer 4 Report

The authors have now submitted a new version of the manuscript, including new information on the burned areas’ mapping. I still think this deserves to be better explained in the manuscript. The questions raised during the second review were poorly answered.

Author Response

Dear Reviewer, thank you very much for your comments. In this version, we present a corrected version of the method for detecting fire scars and although the previous version included the results of the validation of the scars, we decided to include the confusion matrix with omission/commission errors as a new table (Table 3). We hope that these data are sufficient to argue the validity of the results.

This manuscript is a resubmission of an earlier submission. The following is a list of the peer review reports and author responses from that submission.

Round 1

Reviewer 1 Report

The manuscript entitled “Quantification of fire severity and dynamics of post-fire regeneration in the Chiquitania region, Bolivia” presents an interesting case study where remote sensing data and methods are employed to study post fire vegetation dynamics in four study areas affected by wildfires during the period 2019-2021. The main finding of the study is that these ecosystems promptly respond to wildfire impacts by natural regeneration and based on this results the authors make a claim towards promoting natural regeneration, as opposed to artificial planting, and ensuring the protection of the affected areas from further degradation.  Although none of the methods employed is novel, and similar results are reported in other studies I still believe that the study is worth publishing.

However I have some concerns primarily in the way the study is organised and presented. The authors performed multiple analysis using multiple datasets and the results are presented with a combination of text figures and less tables.  The reporting of several numbers in the text reduces the readability of the manuscript and increases the complexity of the reported results. Perhaps the authors should consider tabulating some of the reported results. Also in section 3.4. the authors report some “increases” in minimum NDVI and NBR but, based on the figures, I believe what they mean is “decreases”. Furthermore, I sincerely believe that the most interesting results are reported in sections 3.1 and 3.2. which show a prompt response of vegetation to wildfire when no additional disturbance is involved. The results on the interannual variability of the phenological status of vegetation add little additional information on the study and in my view they are slightly irrelevant, although the authors discuss these changes also in relation to fire and landcover.

In my view the manuscript is interesting but it needs to be a little more focused on the efficiency of Remote Sensing data and methods for monitoring vegetation recovery after fire and on the ecology of these ecosystems which allow them to quickly recover, not of course in terms of species composition and community structure but in terms of green cover, after fire.  I am suggesting major revision not because I have any objection on the scientific soundness of the methods employed but simply because I believe the authors should consider making the manuscript more focused and re-arrange the reporting of the results in a more comprehensive manner. Also, avoid one sentence paragraphs and merge them either with the paragraph before or after.  

Reviewer 2 Report

Dear Authors,

I think the topic you address with your work is of interest and certainly suitable for the selected Journal. However, the manuscript needs substantial revision to better clarify the objectives, data, methods and results. As it is now , unfortunately, it is very difficult to follow, to read and to understand.

All sections (data, methods, results, discussion) are not well organized and mixed up. Please consider to re-write the manuscript and once all sections are fully and well completed, the revision of the scientific contribution could be carried out by a reviewer.

Data and methods sections are poor of information to understand what you have done and what are the characteristics of the data you used. You state that major dataset for your analysis is Sentinel 2 but then you introduce both MODIS and Landsat. This is a real mix up of datasets and if not clearly explained could lead to misunderstanding. Why do you use different datasets? How are these dataset comparable also given the size of the field plots? Where are field plots located? How are data processed? You cannot always refer to other works (especially if the are not easily accessible) but you should provide all information the reader needs to understand what you have done. 

Also results are a list of numbers and figures that could be better presented to help reader to spot the most interesting results. Some of the figures could be presented side by side especially when comparing burned and unburned boxplots to enhance the difference. Moreover, the discussion should fuse all results together, as results are presented now they are a list and it is very difficult to find relationships between the difference results (for example, what is the link between filed plot sampling measurements, CBI) and severity derived from dNBR? I think this is missing and it is not up to the reader to find it out; unless I could not spot it in the text.

Throughout the manuscript several information provided are wrong or not precise. For example NBR is Normalized Burn Ratio and not "burning", what do you mean with regeneration? You state that you "calculate the cumulative annual of burned areas for the period 2001-2020 from MODIS in GEE: how is it used? What are the dates interval you consider? It is very difficult to understand, please may be focus on a subset of all analysis you have done but present everything in a more clear way. These are some examples of the not linear presentation of thew data and analysis carried out in this work.

I'm not mother tongue, but English may need revision. 

For all the above mentioned reasons, I think the manuscript is not suitable for publication and Authors could consider, once analyses and text are fully revised, to submit a new contribution.